# Position: It Is Time We Test Neural Computation In Vitro

**Frithjof Gressmann** [1]   **Ashley Chen** [1]   **Lily Hexuan Xie** [1]   **Nancy M. Amato** [1]   **Lawrence Rauchwerger** [1]

## Abstract

Recent advances in bioengineering have enabled the creation of biological neural networks *in vitro*, significantly reducing the cost, ethical hurdles, and complexity of experimentation with genuine biological neural computation. In this position paper, we argue that this trend offers a unique and timely opportunity to put our understanding of neural computation to the test. By designing artificial neural networks that can interact and control living neural systems, it is becoming possible to validate computational models beyond simulation and gain empirical insights to help unlock more robust and energy-efficient next-generation AI systems. We provide an overview of key technologies, challenges, and principles behind this development and describe strategies and opportunities for novel machine learning research in this emerging field. We also discuss implications and fundamental questions that could be answered as this technology advances, exemplifying the longer-term impact of increasingly sophisticated in vitro neural networks.

## 1. Introduction

In recent years, two mutually influential fields, artificial intelligence (AI) and neuroscience, have witnessed revolutionary developments. The remarkable success of large-scale neural networks in machine learning (ML) has enabled the effective modeling of complex data patterns and relationships across a wide variety of domains (Bommasani et al., 2022). In bioengineering, groundbreaking work on induced pluripotent stem cell (iPSC) technology has enabled the conversion of ordinary cells to stem cells (Takahashi et al., 2007), facilitating the in vitro cultivation of neural cell cultures for study and application outside of their nat-

ural biological context (Smirnova et al., 2023). Together, these advances present, as we will argue, a novel synergistic opportunity. Historically, the neuroscientific study of the brain has served as a source of inspiration in developing artificially intelligent systems. As neuroscientific study sets out to *uncover* working principles of perception, behavior, learning, memory, and reasoning (Finger, 1994), AI research made progress *engineering* such capabilities in silicon-based artificial systems. With the advancement of both iPSC technology and large-scale data-driven ML, it is now possible to consider an intersection of these efforts, namely a "reverse engineering" of neural computation using living cells. Concretely, this means developing enough practical understanding of cultured in vitro neurons[1] so that they can be modeled and controlled to elicit or reproduce cognitive abilities much like their artificial deep learning counterparts. Notably, we construe neural computation as systematic transformation of information encoded in neural activity patterns that can be functionally characterized as a task performed by the in vitro system (see Figure 1).

### 1.1. In vitro neural networks as a testbed for neural computation models

The idea of reverse engineering biological neural processing to abstract the underlying principles is, of course, not new and presents a long-standing priority of NeuroAI (Zador et al., 2023). However, we contend that the engineering and experimental interaction with *stem cell-derived* neurons presents a novel and unique opportunity for at least three reasons. First, although in vitro cell cultures recapitulate many structural and functional aspects of brain tissue, they are simpler and much smaller, reducing the complexity of the system (Zhao et al., 2022). Secondly, in vitro experiments greatly enhance the possibilities of causal intervention to validate experimental hypotheses about the system (Takebe & Wells, 2019). Specifically, it is possible to control environmental conditions, repeat procedures with multiple cell cultures, and manipulate and observe in ways that would be impossible or unethical with in vivo subjects. Finally, the technology has matured to the point where open

---

[1]Siebel School of Computing and Data Science, University of Illinois at Urbana-Champaign, Urbana, IL, USA. Correspondence to: Frithjof Gressmann <fg14@illinois.edu>, Lawrence Rauchwerger <rwerger@illinois.edu>.

*Proceedings of the 42$^{nd}$ International Conference on Machine Learning*, Vancouver, Canada. PMLR 267, 2025. Copyright 2025 by the author(s).

---

[1]Throughout this paper, *in vitro neurons* refers specifically to neurons derived from induced pluripotent stem cells (iPSCs), not neurons obtained from other sources such as primary cultures or neuroblastoma cell lines.

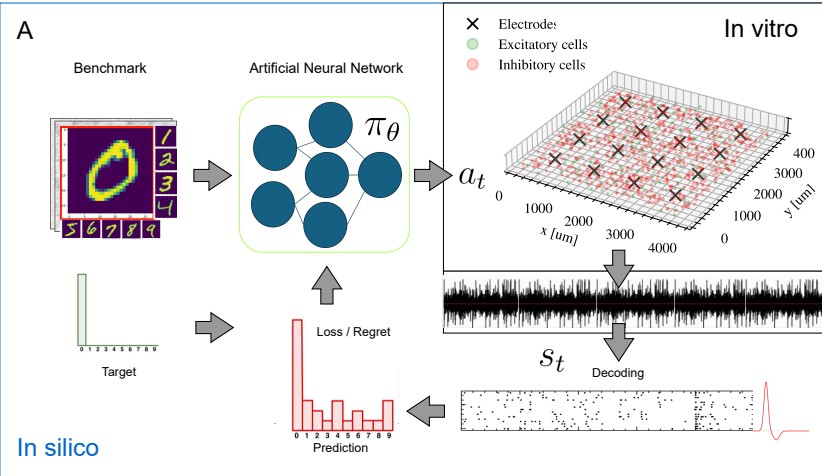
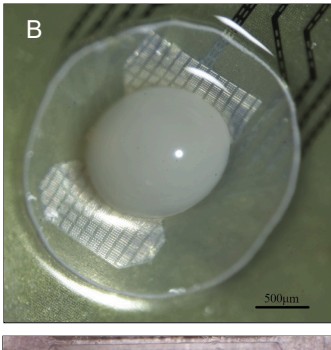
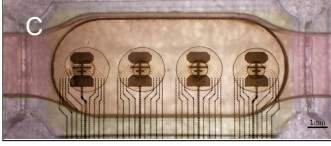

*Figure 1.* (A) Schematic overview of a possible in vitro closed-loop setup where an artificial neural network can manipulate and observe in vitro neurons via a multiple electrode array (MEA) or possibly optical stimulation. The goal is to interact with the living system so it solves a given benchmark task. (B) Microscope view of a commercial experimental in vitro system by Jordan et al. (2024) mounted atop an MEA that is shown from below in (C). [Photos by Jordan et al. (2024)/CC BY 4.0]

and cost-effective platforms are becoming available more broadly (Zhang et al., 2024). In particular, in vitro "wetware" can now be made available in cloud computing-like arrangements, allowing ML researchers without lab training to experiment with the systems (Elliott et al., 2023; Jordan et al., 2024). These developments raise the prospect of novel, unconventional experimental platforms that can enable machine learning research to extend beyond the common purely simulation-based studies. Notably, in vitro neural networks should be of special interest to machine learning researchers at the intersection to neuroscience and neuro-inspired AI, including Spiking Neural Networks (SNNs) and neuromorphic computing. While work on biologically plausible computational models has soared in recent years (Fang et al., 2023, Figure S28), it has largely been confined to a conventional ML problem domain that – at least for now – is best solved by standard artificial neural network approaches. In last year's open review of ICLR, an anonymous reviewer acknowledged the growing community interest in spiking neural networks but cautioned that "[v]ery rarely does an SNN paper show its advantages in the broader literature on neural networks, let alone in the real world" (Hammouamri et al., 2023). In vitro cultures thus present a timely frontier for moving beyond today's silicon architectures with important applications ranging from medical treatments of neurological diseases over bio-robotics to brain-computer interfaces. Given these opportunities, we argue that ***the time is right to pursue the development of ML models that learn to interact with biological neural networks to test and advance our understanding of both biological and artificial neural computation***.

## 1.2. Demonstrating understanding by engineering useful living neural systems

Despite its promising prospects, the field of engineering cell cultures for computing applications remains in its infancy. To illustrate this, consider that the engineering of conventional von Neumann-type computers was driven by a theory of computation developed before any real-world prototypes emerged. The engineering problem was not how a Turing machine-like device could compute in theory but rather how to implement such a system in practice. When it comes to the development of biological tissue for computing, however, two problems arise: figuring out the *practical* challenges of growing cells from stem cells (i.e. the bioengineering) and, at the same time, developing a *theory* of how what has been developed works or does not work (i.e. the neuroscientific reverse engineering). This has important methodological implications. In particular, data collection and algorithmic analysis of in vitro neural activity in itself may be of limited value as long as a formal framework for its interpretation is lacking (Lazebnik, 2004). Jonas & Kording (2017) illustrated this by attempting to reverse engineer the known working principles of a conventional microprocessor using neuroscientific data analysis methods alone. While the data-driven models uncovered structure, they ultimately did not elucidate the actual hierarchy of information processing in the microprocessor. This highlights a key issue in data-driven neuroscience, namely that effective modeling of observed data does not guarantee meaningful insights as long as alternative models and explanations remain equally plausible. In other words, there needs to be a method of arbitration for features of the data that are indicative of facts

about the system as opposed to spuriostus complexity irrelevant to the phenomena (Boyd & Bogen, 2021). We contend that learning to interact with in vitro systems provides a practical method of arbitration because the modeling quality can be judged as a function of the resulting control of the system. For instance, a controller model trained on observable activity can be assessed by its capability to steer the neural activity toward desired states. This is reminiscent of an embodied Turing test (Zador et al., 2023) where the goal is to achieve command of real-world environments. By framing the problem as a control problem amenable to optimization, data collection becomes goal-directed despite the limited understanding of the principles that underpin the data-generating process. Notably, in practice, implicit learning of control to leverage neural systems is seeing growing success. For example, brain-computer interfaces tested with human patients have been demonstrated to decode thought from neural activity recordings with remarkable precision (Gao et al., 2021). For simpler organisms such as the Caenorhabditis elegans nematode, optogenetic stimulation has been used to induce basic motor control (Li et al., 2024). A key ingredient in these achievements has been effective machine learning methods that can build rich, implicit representations of the observed system dynamics to solve downstream problems. However, the high cost of in vivo experimentation and data collection is likely a limiting factor to the pace of machine learning innovation in the field. Thus, in vitro platforms present a timely opportunity to broaden access and increase the pace of cross-pollination between neuroscience and machine learning innovation. Ideally, these trends bootstrap a virtuous cycle of discovery where an increased capability to infer and model principles of living neurons translates to improved coding and control in the experimental system. This would, in turn, lend itself to validating new hypotheses about the functioning of neural dynamics and further improve the understanding of neural information processing.

### 1.3. Potential for the development of next-generation AI systems

In the long term, these efforts could provide a pathway for leveraging biological neural networks and contribute to our understanding of what makes biological learning in neurons so incredibly efficient, holding broader lessons for the development of more energy-efficient AI systems. Contemporary machine-learning strategies still struggle to deal with the uncertainty and overwhelming complexity of the real world. Thus, a shift from conventional digital to cellular substrates presents a uniquely challenging benchmark that could help address blind spots in engineering practices and real-world performance of state-of-the-art models (Herrmann et al., 2024). It is reasonable to expect that the shift from exactness, high accuracy, and reliability to statisti-

cal and potentially unreliable processes will make for a useful modeling test case that could help drive progress for more robust machine learning models and algorithms. Furthermore, while experiments with in vitro neurons will not necessarily yield neuroscientific insights into the brain, ML-driven neuro-integrated platforms could help open the door to transformative insights into human biology, disease modeling, and drug discovery, offering significant medical advancements and societal benefits (le Feber, 2019; Kropp et al., 2017; Kim et al., 2020).

The remainder of this paper is organized as follows. Section 2 provides a brief primer on key principles, technologies, and challenges of engineering in vitro neural networks. Although our review is far from comprehensive, it provides pointers to surveys in the respective disciplines that shape this emerging field. Against this background, in Section 3, we describe possible modeling and optimization strategies to learn to harness in vitro systems and discuss open challenges and future directions. We also consider alternative views and critiques of our position in Section 4. Finally, we outline the implications and impact of continued progress on this frontier in Section 5, before concluding in Section 6.

## 2. Principles, technologies, and challenges

### 2.1. Biological neural networks

Given the significant success of neural networks in machine learning, it can be easy to forget that *artificial* neurons are a radical simplification of their biological counterparts that originally inspired them. Biological neurons do not only encode information in a fundamentally different way through spiking temporal dynamics (Roy et al., 2019), but also leverage processes such as synaptic, homeostatic and structural plasticity (Billaudelle et al., 2021; Yang et al., 2020), local error propagation via dendritic computation (Pagkalos et al., 2023; Liu et al., 2023), or neuromodulation (Cheong et al., 2022) whose complexity far exceeds those of artificial neurons.

At a lower level, however, the general physiological principles that give rise to the complex neural dynamics have been uncovered. Put simply, biological neurons transmit signals in the form of potential differences between ions that are separated by the cell membrane (Ekeberg et al., 1991). The opening of ion channels in the membrane causes the cell to depolarize, a process that propagates along the membrane toward downstream cells (see Figure 2). Upon reaching the synaptic terminal, neurotransmitters diffuse to and induce a current in the post-synaptic neuron, which continues the signal transmission chain.

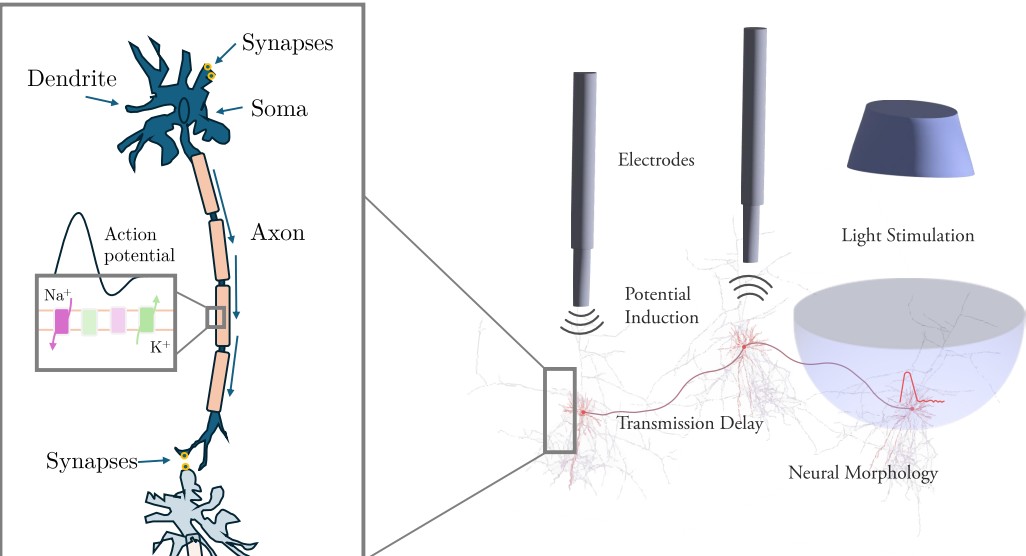

*Figure 2.* (Left) Illustration of the anatomy of a single neuron where inputs reach dendritic synapses to ultimately depolarize the cell and transmit an output to the terminal synapses that continue the transmission chain. Action potentials are generated when depolarization opens voltage-gated sodium channels, allowing $Na^+$ ions to rush into the cell, and are transmitted along the axon as this depolarization triggers adjacent voltage-gated channels to open in sequence, followed by potassium channels opening to restore the resting potential through $K^+$ efflux. (Right) Illustration of common types of experimental interaction with biological neurons. Electrodes can change the extra-cellular potential to illicit depolarization and thus induce spiking activity. At the same time, they can pick up spikes from nearby neurons. Alternatively, optical stimulus can precisely activate or deactivate special ion channels that have been introduced into specific cells. The neural morphology and cell connectivity influence the interaction and resulting dynamics.

## 2.2. In vitro neural networks

Although the field has historically focused on studying biological neurons in vivo, advances in bioengineering are popularizing the use of in vitro technologies and allowing the development of in vitro neural networks for computation.

The key to this development is induced pluripotent stem cells. They are a type of pluripotent cell derived from adult somatic cells that have been reprogrammed to an embryonic-like state, providing a virtually unlimited and less ethically problematic source of cells for biomedical research (Takahashi et al., 2007). In particular, the so-called "organoid" technology is driving further progress to enable increasingly sophisticated in vitro applications (Zhao et al., 2022). Organoids are artificially generated three-dimensional (3D) cultures of cells derived from iPS cells. They can contain cell types that self-organize through cell-sorting processes and spatial restrictions. Researchers often opt for 3D cultures over 2D cultures to obtain more physiologically realistic cellular compositions and achieve extensive culture growth while maintaining the potential for high-throughput screenings and analysis (van de Wetering et al., 2015). For example, high-content imaging (HCI) and machine learning strategies allow a fast analysis of organoid data (Costamagna et al., 2021).

## 2.3. Neural recording and stimulation

Multiple physiological processes can be leveraged to interact with and manipulate the neural dynamics of these in vitro systems (Yang et al., 2024). For one, changing the extracellular potential within neurons can illicit depolarization and thus induce spiking activity. Alternatively, manipulation of the ion-channel permeability can alter the current flow and neural processing as a result. More fundamentally, neurotransmitter and blocker agents can interfere with the chemical balance at the synapses and modulate the synaptic neurotransmission. In practice, experimental techniques leverage these principles to establish control over neural dynamics (see Figure 2).

**Multi-electrode arrays**   As one of the most established neuromodulation techniques (Rey et al., 2015; Thornton et al., 2019), electrical stimulation is commonly realized with extra-cellular electrodes that can detect and deliver potential differences in surrounding cells (Ronchi et al., 2019). In particular, multi-electrode arrays (MEAs) that arrange electrodes in configurable mesh-like layouts allow high-resolution electrophysiological measurements with minimal disruption to cell tissues (Chen et al., 2017). However, a major limitation of electrical stimulation is its inability to target specific cells and regions due to the spread of current (Won et al., 2020).

**Optogenetic stimulation** Optogenetics has emerged as a promising alternative for neurostimulation, as it uses light to manipulate specific neurons and neuron groups (Deisseroth, 2011; Xu et al., 2023; Chen et al., 2022). The method involves introducing foreign light-sensitive transmembrane proteins, known as opsins, into target cell populations (Montagni et al., 2019). Opsins may, for instance, be delivered via viral infection, allowing the targeting of specific cells (Yizhar et al., 2011). Subsequent light stimulation can then precisely activate or deactivate ion channels and neuronal activity without affecting neighboring cells. In particular, there is a wide variety of different microbial and genetically modified opsins that allow flexible experimental design (Masseck, 2018). For example, certain opsins may respond to low-intensity light, minimizing potential cell damage (Rodgers et al., 2021). Optogenetic stimulation also works well with MEA-based systems, allowing for increasingly integrated experimentation platforms (Shin et al., 2021; Brosch et al., 2020; Welkenhuysen et al., 2016). Thus, it is no surprise that optogenetic stimulation is seeing widespread adoption for in vitro experimentation (Zabolocki et al., 2020; Morton et al., 2019; Hallett et al., 2016; Zhang et al., 2024; Xu et al., 2023; Chen et al., 2022).

### 2.4. Neural coding and data processing

With the technology to create, manage, and record neurons in vitro in place, it becomes essential to effectively decode information from the observed raw neural activity, a process that poses various challenges. Modern recording devices, such as MEAs, allow the simultaneous recording of activity from hundreds to thousands of neurons (Hurwitz et al., 2021), making neural data extremely high dimensional. Neural processes are also inherently stochastic. Synaptic vesicles, for example, are known to spontaneously release neurotransmitters even in the absence of evoked activity, causing random activity fluctuations as a result (Andreae & Burrone, 2018). At the same time, recording devices and techniques introduce additional noise and uncertainty. For instance, since MEAs typically record extracellularly from multiple cells, complex post-processing algorithms that determine which neurons fired are common, adding another layer of uncertainty (Garcia et al., 2022).

Several neural coding strategies, including rate, temporal, rank, and direct coding, have been proposed to extract and represent the information content of neural activity (Taherkhani et al., 2020; Yi et al., 2023). However, it remains unclear to what degree biological neural networks actually employ such encoding schemes. Furthermore, recent work suggests that neural activity may be effectively represented in fewer dimensions, indicating that the high-dimensional nature of neural data might be highly redundant (Idesis et al., 2023). However, identifying these low-dimensional representations within the highly non-linear

population activity remains a challenge (Fortunato et al., 2024). Many widely used dimensionality reduction techniques, such as Principal Component Analysis (PCA), make linear assumptions and may not capture data patterns effectively. At the same time, more sophisticated, non-linear dimensionality reduction methods, such as autoencoders, often struggle with issues such as noise and overfitting in neural data (Altan et al., 2021).

## 3. Learning to control in vitro neural networks

As the technological foundation of in vitro neural networks is being established, seizing on its potential will require the development of new machine-learning approaches that can process the vast observable activity of neuronal cell cultures and learn to make sense of their neural code. While many approaches have been proposed and explored, here, we advocate for a focus on learning to interact with and control the system by reacting to observed neural activity response. Notably, Kagan et al. (2022) provided a proof-of-concept of such a closed-loop environment interaction where the activity feedback of in vitro neurons was used to realize basic video game play. While the training method did not leverage explicit gradient computation but relied on heuristics, it is straightforward to generalize this setting as a reinforcement learning (RL) problem with a stochastic environment so that common training strategies of model-free and model-based RL apply. Li et al. (2024), for instance, used a Soft Actor-Critic (Haarnoja et al., 2019) algorithm trained offline on prerecorded data to discover viable neural control policies. In this section, we discuss such possible strategies for "in vitro training" and highlight open problems as opportunities for future research.

### 3.1. Problem formulation

While the technology and capabilities of experimental systems can vary significantly (Section 2), at a basic level, controlling in vitro cell cultures comes down to figuring out a stimulation sequence in response to observed activity. Specifically, a control model needs to learn to predict appropriate *stimulation* of the available input channels at certain *times*. This may be, for instance, a sequence of times when to deliver stimulation through certain electrodes or via laser-induced optogenetic means (cp. Figure 2).

More formally, the in vitro neural network can be characterized as an unknown stochastic transition function $\text{IVN}(s_{t+1} \mid s_t, a_t)$ where $s_t \in \mathcal{S}$ represents the observed neural dynamics state and $a_t \in \mathcal{A}$ the stimulation input at time $t$. The optimization objective is to find a set of parameters $\theta$ such that the control policy $\pi_\theta : \mathcal{S} \to \mathcal{A}$ steers the observable neural dynamics of the biological neural network system in some desirable way. Note that this formulation does not assume anything about the internal

characteristics of the neural systems. Training $\pi_\theta$ successfully means not only overcoming the practical challenges of controlling a noisy, complex system, but it also implies uncovering some properties of the IVN that can be exploited to achieve the given objective. For instance, the policy could simply use the IVN as a random projection into a higher dimensional space (this would be reminiscent of reservoir computing, see Maass et al. (2002); Lukoševičius & Jaeger (2009); Tanaka et al. (2019); Cucchi et al. (2022)). A more sophisticated model, however, may learn to exploit more intricate properties of the IVN. For example, the model may leverage present plasticity by repeatedly delivering simulations to reconfigure the synaptic connectivity of the network. Crucially, given a suitable optimization approach, the discovery of control strategies can imply an implicit discovery of the working principles of the biological system and thus progress towards the ultimate goal of neural reverse engineering.

### 3.2. Optimization approach

How to find an effective policy $\pi_\theta$ for in vitro systems is, in general, as much of an open question as what model and training approach would be most suitable. There are, however, principles that can guide the experimentation.

First, the relatively high cost and unreliability of lab environments compared with simulators mean that the training of $\pi_\theta$ will likely rely on a pretraining scheme using synthetic or pre-recorded data with subsequent fine-tuning on the more limited real-world data. Notably, the long-standing developments in high-fidelity neural simulation present a rich resource for generating realistic synthetic data of neural dynamics. Thus, developing a simulation-driven pretraining corpus for a large-scale policy sequence model $\pi_\theta$ is likely a worthwhile first step. One key question in this effort will be what level of simulation fidelity is required to allow $\pi_\theta$ to represent relevant neural dynamics without over-fitting. Evidence from real-world data suggests that pre-trained representations may be able to bridge considerable transfer gaps. For instance, it has been demonstrated that pre-trained representations of neural activity can be general enough to transfer to different data domains, for example, between muscular electromyographic (EMG) signals to electroencephalographic (EEG) brain activity (Bird et al., 2020). Furthermore, as likelihood-ratio gradients illustrate (Williams, 1992), it can be more important to accurately simulate the observable high-level system response than faithfully replicate all the underlying intricate neural dynamics. To illustrate this, consider a basketball player who does not need to understand physics equations of ball trajectories to improve shots but focuses on whether the ball goes in. It may thus be sufficient to generate and train on synthetic data that only loosely match the lab data encountered at fine-tuning and inference time.

With suitable and sufficient data in place, the question becomes how to optimize $\pi_\theta$. Model-free RL approaches are directly applicable but may be challenging due to the relatively limited time spent interacting with the living culture. It is thus essential to improve modeling and gradient estimation strategies of neural systems that could boost more sample-efficient model-based optimization. While non-continuous spiking dynamics are not differentiable in general, work on spiking neural networks (SNNs) has brought about a wide range of applicable optimization techniques (Roy et al., 2019; Tavanaei et al., 2019; Zenke & Vogels, 2021). In particular, surrogate gradient techniques offer a straightforward way to apply backpropagation-driven training to otherwise non-differentiable spiking dynamics (Neftci et al., 2019). For the leaky-integrate and fire neuron model, several methods (Bohte & Kok, 2000; Booij & tat Nguyen, 2005; Xu et al., 2013) provide exact gradients and can implement event-based gradient computation within the dynamical system (Wunderlich & Pehle, 2021; Holberg & Salvi, 2024). Moreover, recent work has introduced methods for differentiable simulation of detailed biophysical models (Deistler et al., 2024). These advances allow for the development of differentiable neural simulations that integrate the power of backpropagation-based machine learning models with theoretical and experimental models of biological neural networks (Richards et al., 2019). Much like the deep learning framework's auto-differentiation supported the rise of artificial neural networks (ANNs), it is conceivable that automatic gradient computation for biological models will greatly accelerate progress in the field.

Another crucial challenge is the design of suitable objectives or rewards. First, it is worth stressing that the ultimate goal is to find a policy that finds ways to *exploit* the biological neural network in the quest of minimizing the loss or regret. In other words, the IVN should be seen as an extension of the artificial policy network since it is itself a "trainable" neural network, albeit with very different and currently poorly understood learning rules. Thus, the optimization objective should incentivize the discovery of the rules that allow the policy to leverage the IVN in non-trial ways. In fact, ideally, as $t \rightarrow \infty$, the artificial policy should approximate the identity function, meaning that it has interacted with and reconfigured the biological system in such a way that the IVN itself now minimizes the regret. This is reminiscent of a teacher-student (Hu et al., 2022) or a knowledge distillation setting (Gou et al., 2021) with a biological student, and work in this area may provide lessons for effective reward design.

Naturally, the difficulty of problem benchmarks to solve using the IVN will be modest initially and gradually increase as the technological and modeling capabilities progress. This means that the creation of standardized benchmarks at gradually varying difficulty levels represents an important problem in itself. Many problem types that drove the de-

velopment of ANN research, such as pattern classification, gameplay, or sequence prediction, may turn out to be useful in vitro benchmarks. At the same time, just as the dataset sizes increase with available computing capabilities, it will be important to adjust the in vitro benchmark to match the available level of experimental sophistication, particularly concerning the available stimulation and measurement capability. For example, assuming a presentation time of 500 ms per image, a single epoch of MNIST training would require almost seven hours of recording time, likely exhausting contemporary lab settings where typical recording timelines range over a few hours to days (Zhang et al., 2024). Furthermore, a task such as classifying digits may not be best to incentivize the implicit goal of discovering how to control the neural system. After all, task-solving abilities are, first and foremost, a measure of system control rather than the end goal (although, admittedly, in vitro MNIST inference on analog wetware would represent an impressive feat).

### 3.3. Application in silico and in vitro

Much like in other real-world interfacing fields, such as robotics, it is likely that a significant part of the research and development will be simulation-driven. Notably, simulation environments such as Cleo (Johnsen et al., 2023) allow for low-stake testing and debugging of closed-loop experimental setups to inform eventual lab experiments. However, these tools have considerable sim-to-real gaps limiting their applicability. Thus, more work is needed to improve simulation environments of in vitro systems while narrowing the gap to real-world deployment. While simulations can provide a development test-bed, ultimately, optimization strategies must be tested on actual in vitro platforms. In the simplest case, this may be a pre-trained policy that is rolled out against the in vitro system without taking the current system state into account. Naturally, such offline, open-loop strategies will have limited capabilities as they cannot react to specifics of ever-evolving in vitro dynamics. Closed-loop training setups where the policy receives currently observed system states to feed back an action can be more powerful but are harder to realize on a technical level. Since typical neural dynamics play out on a millisecond time scale, the policy inference must meet latency constraints, ideally in the sub-millisecond range. This includes data transfers, recording, and pre-processing of the current neural activity, for which the raw data volume can be substantial. For instance, common MEAs sample up to hundreds of channels, yielding tens of kilobytes of data per second and channel. Thus, to enable closed-loop pipelines, non-trivial inference systems, such as custom FPGAs that can directly interface with the electrode recording system and accelerate signal processing in hardware, will be required. As such, there are many opportunities to use the valuable lessons and expertise of the MLSys and ML-edge computing community.

More generally, we believe that continued efforts in developing models and optimization strategies, guided by feedback from real-world experiments, can help pave the way to increasingly sophisticated computing applications in vitro.

## 4. Alternative Views

Having laid out our view of the technology, open challenges, and promising avenues of exploration, we consider alternative positions here.

### 4.1. Study of in vitro neurons is unlikely to contribute to AI innovation and development

The remarkable progress of AI development in recent years has arguably led to an increasing divergence of modern artificial systems from their inspirational neuroscientific origins (Editorial, 2024). Despite tremendous efforts in neuroscience, hard-won empirical insights into biological information processing had little influence on the engineering-minded advancements in ML, such as the development of attention-based transformer architectures. In fact, although AI methods were initially inspired by neuroscience (Hassabis et al., 2017), current state-of-the-art models are far from biologically plausible. Modern architectures rely exclusively on backpropagation, which appears to have minimal biological basis (Macpherson et al., 2021). Additionally, smaller-scale, engineered in vitro systems may not provide the kind of insights that could be obtained from brain research with in vivo subjects. While in vitro neurons can establish connections and generate electrical activity patterns similar to those seen in developing brains (Trujillo et al., 2019), they fail to form more advanced synaptic circuits and remain unable to exhibit complex brain functions (Lee et al., 2020; Smirnova & Hartung, 2024). This limitation is especially concerning considering our understanding of the brain and intelligence is still incomplete (Roland, 2023). Consider, for example, that experiments with underpowered single-layer neural networks misled early researchers about the potential of neural networks (Minsky & Papert, 1972). Moreover, shortcomings of in vitro systems can be difficult to identify and, consequently, challenging to correct with existing validation strategies (Dauth et al., 2017; Smirnova & Hartung, 2024).

However, this engineering-first perspective may be overly dismissive of the areas where biological systems excel, such as energy efficiency, continual learning, and robust generalization. Even if in vitro systems cannot perfectly replicate brain function, they provide the kind of controlled environment necessary for research that can shed light on these relevant issues. The historical divergence between neuroscience and AI development should not preclude future convergence, especially as both fields continue to mature and evolve.

### 4.2. The technology is not ready to build atop

As previously discussed, working with in vitro neurons poses many practical difficulties. Even with a functional in vitro platform in place, maintaining healthy and functioning neurons is challenging as these neurons often suffer from stress, hypoxia, and necrosis (Kim & Chang, 2023; Li et al., 2023). This means that hard-to-scale expertise is still required to offer broader access to in vitro computing systems. Additionally, the intersection of in vitro research and AI is often driven by objectives related to improving human health and understanding the brain in that context (Zador et al., 2023). This means that a lot of research efforts, funding, and institutional support are directed toward medical applications, leaving less room for initiatives that explore how neuroscience could inform AI development. The lack of coordinated effort and institutional backing can lead to fragmented interdisciplinary collaboration, with results that are often sporadic and disconnected. Notably, entering this field without the necessary resources carries significant risks since progress will likely be slow, unstructured, and prone to failure.

Nevertheless, these technical and institutional challenges, while significant, may be temporary rather than fundamental barriers. The rapid advancement of bioengineering technologies and robotics continues to make neuronal maintenance more reliable and accessible. Importantly, in vitro "cloud providers" could absorb the maintenance burden and complexity and unlock scaling benefits much like conventional cloud infrastructure providers (Jordan et al., 2024). As with many emerging technologies, initial difficulties in coordination and resources may simply represent the field's nascent stage rather than insurmountable obstacles to progress.

### 4.3. It is unethical to engineer living neurons

Work with living organisms and stem-cell technology has to consider ethical implications. At this time, it is not standard practice to inform cell donors that their tissue could be used for iPS cell derivation or to generate neural systems based on their genetic material (Hyun et al., 2020). Given that skin cells, saliva, or small amounts of blood are typically all it takes, donors may not be aware of potential uses of their tissue (Kagan et al., 2023). Without full transparency about the risks and benefits of their donation, ethical concerns and moral or legal blind spots will only grow. Furthermore, one potential longer term risk is that in vitro neurons could develop a form of consciousness, making them capable of suffering (Hyun et al., 2020). Such a development, while unlikely in the near future (Milford et al., 2023), raises important questions concerning welfare and the ethical limits to using such systems for research. Currently, there are very few, if any, regulations concerning in vitro systems to guard against or handle such a development (Goddard et al., 2023).

Additionally, it is difficult to ascertain consciousness due to a lack of universally agreed-upon definition and the absence of reliable tests to determine its presence (Reardon, 2020).

While these ethical concerns deserve serious consideration, they need not entirely preclude research in this field. The issues of donor consent can be addressed through improved informed consent protocols and transparent communication about potential applications. Furthermore, at the current scale and complexity, in vitro neural systems remain far from anything that could plausibly support consciousness (Milford et al., 2023), and careful regulatory frameworks could be developed proactively to establish ethical boundaries as the technology advances. Many emerging technologies have faced similar ethical challenges and developed robust governance frameworks to ensure responsible development. Rather than abandoning the field, the focus should be on establishing clear ethical guidelines and oversight mechanisms while the technology is still in its early stages.

## 5. Longer-term implications

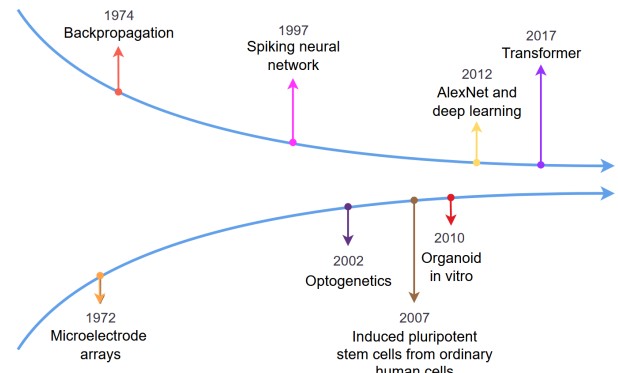

*Figure 3*. Historic timeline of converging advances in machine learning and bioengineering using induced pluripotent stem cell (iPSC) technology. We argue that the technology has matured to the point where it enables novel, converging experimental approaches to interrogating neural computation beyond purely simulation based studies. Future efforts in this field may contribute to uncovering the working principles of biological neural processing, particularly in regard to their remarkable energy efficiency and robustness to noise and uncertainty. As such, it stand to reason that engineered living biological networks may ultimately pave the way for next-generation hardware for artificial intelligence applications, be it in silico, in vitro, or both.

Engineering in vitro neural networks represents a very nascent undertaking. However, it is not too early to consider its longer-term potential and implications as the technology advances (see Figure 3).

AI has already profoundly impacted open problems in the biological sciences, including protein folding (Jumper et al., 2021), drug discovery (Wallach et al., 2015), and evolutionary biology (Sheehan & Song, 2016). Similarly, it is possible that ML-driven research with in vitro neurons could contribute to significant conceptual advances on major open problems in the field. For example, one of the key questions with significant implications for AI and neuroscience concerns the energy efficiency of neural systems. The huge divergence in power requirements between artificial and biological neural networks suggests a greater energy efficiency of biological neural learning. In fact, living neurons are known to operate at extreme levels of sparsity and tolerance to noise, but exactly how it is achieved remains unclear (Zador, 2024). A reverse-engineering of energy-efficient and data-efficient neural processing could lead to architectures with much lower power requirements and mitigate the enormous energy requirements of the recent AI boom (Bojic et al., 2024).

At the same time, energy efficiency may turn out to be a direct consequence of one of the most notable characteristics of biological systems, namely their mortality. As Hinton (2022) pointed out, today's digital computers have a clear separation between the hardware and software layer. This makes it possible to copy software or model weights to a new hardware substrate, making the software "immortal". However, achieving such portability requires, by definition, a layer of abstraction, which costs energy. Thus, "[if] we are willing to abandon immortality, it should be possible to achieve huge savings in the energy required to perform a computation and in the cost of fabricating the hardware that executes the computation" (Hinton, 2022, p. 13). Biological neural networks may be operating on this principle as it is not possible to separate weights from the biological substrate, and the "software" thus dies with it. Arguably, in vitro systems offer a plausible route to investigating this idea that may help realize more energy-efficient, abstraction-less systems.

More generally, the mortality of the in vitro system has important implications. Since the system state can never be check-pointed or exactly reproduced, a form of communication will be the only way to teach or read out what has been learned. In this context, work to freeze and thaw the cultures may offer insights into the relationship between the frozen structural properties of the cells and the alive dynamics of neural information processing (Whaley et al., 2021). This line of research may also be relevant to AI safety debates where some proposed safety strategies, such as neuromorphic AGI or mammalian value priors, crucially rely on insights from the brain (Everitt et al., 2018, pp. 14).

Finally, looking back at the deep learning revolution, it is worth noting that early conceptual advances did not reveal their potential more broadly until sufficiently fast computing and large enough datasets catalyzed AlexNet to win the ImageNet competition. It is possible that progress at the intersection of neuroscience and AI is still in the pre-AlexNet phase. Li et al. (2024), for instance, noted that "[i]t was infeasible to collect thousands of hours of recordings in our environment, and [...] adequate computer simulations of the C. elegans nervous system and its behaviours are not available to generate training data" (p. 727). With the entrance of cheap, flexible, in vitro platforms, paired with increasing ML capabilities, it is plausible that we might be entering a new phase of innovation whose long-term implications may extend even beyond our current capacity to imagine or predict them.

# 6. Conclusion

We have reviewed an emerging interdisciplinary endeavor to develop the technology to harness *in vitro* biological neural networks for computing applications. We argue that a key to this effort should be the development of machine learning models that can learn to control in vitro systems. In this framing, the interaction with the in vitro system can be formulated as an optimization problem to solve a concrete task, leaving the uncovering of the system's working principles as an *implicit* goal. Given the proven ability of machine learning methods to implicitly learn rich representations from data-driven optimization problems, it may become possible to effectively interact with living neural networks despite our limited understanding and experimental control of the underlying dynamics. Besides the direct practical motivations, continued progress in this field is likely to help uncover the working principles of biological neural processing, which could have potentially profound implications for both ML and neuroscience. This synergistic approach, combining biological neural networks with artificial intelligence, may not only advance computing capabilities but also illuminate fundamental principles of neural computation that have evolved over millions of years.

## Acknowledgements

We are grateful to the three anonymous reviewers whose constructive feedback helped clarify and strengthen key arguments in this paper. The work was funded by NSF Expedition "Mind in Vitro" award #IIS–2123781.

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
