# OpenReview forum: "Position: It Is Time We Test Neural Computation In Vitro"
_ICML.cc/2025/Position_Paper_Track — ICML 2025 Position Paper Track poster_

### Official Review · Reviewer_fXVn · 2025-03-05

**Significance:** 4
**Argument Clarity:** 4
**Rating:** 4
**Confidence:** 5

**Questions:**

See above

**Discussion Potential:**

3

**Paper Summary:**

The paper proposes a position to call for actions about testing neural intelligence in vitro by leveraging cutting-edge neural bioengineering techniques. In vitro neural models provides a unique way for researchers to study the root of energy efficiency, robustness and learning mechanisms of biological neural circuits. The paper introducs the background of in vitro neural systems, recoding and manipulation techniques, neural coding and data processing strategies, problem formulation, thoughts about practical optimization approaches. Opposite views are also discussed, where the authors argues the limit of network scale and in vitro technology should not hamper the importance of testing neural computation in vitro. Ethical issues are also discussed.

**Position:**

Yes

**Position In Title:**

Yes

**Related Work:**

4

**Strengths And Weaknesses:**

I like this paper since it throws out a position that looks suprising at the begining but it becomes increasingly reasonable after carefully thinking. As there has already been a proof of concept from Kagan et al. 2022, the in vitro neural computation methodology is at its early stage. I especially appreciate the authors' idea about implementing spiking neural networks (SNN) in biological "hardwares", which I also believe will excite discussion to SNN researchers.  The idea about in vitro cloud provider (Sec.4.2) is also interesting.

The entire paper is well-written, with abundant reference to important related work. I could easily follow this paper as I have some background on the required neuroscience knowledge. The core position is well established, with fruitful discussion about the motivation, sugnificance, practical challenge, and long-term implications. I would agree that the paper will contribute to interesting discussions among the deep learning researchers as well as neuroscientists.

However, I believe that the paper could be improve in several ways. First, having more explanation of the key concepts and technology in neural bioengineering would be helpful to the audience from ML community (which is the majority of ICML attendees). For example, the basic components of the biological neural system (soma, dendrite, synapse...) might not be familier to all the audience. Second, I feel that Sec. 3.2 could be more concise, and more space could be left to the authors' expected timeline (at the scale of 5-50 years) of the development of in vitro neural intelligence. Moreover, additional nice illustrative figures could help the readers to better understand the key ideas.



Some Minor problems:
- line 341: (noa, 2024)
- Reference: Hinton 2023: miss URL or publisher
- line 832 left: miss URL
- line 797 right: miss URL

**Support:**

3

---

> ### Author Rebuttal · Authors · 2025-04-01
>
> Thank you for your thoughtful comments and feedback! We are pleased that you found our position surprising at first, but reasonable after carefully thinking.
>
> We greatly appreciate your suggestions for improvement.
>
> We will expand the presentation of key concepts and technology in neural bioengineering and expand the illustrative figure content, such as Figure 2, to make the article more accessible to its intended audience.
>
> We agree that Section 3.2 can be shortened. Giving more room to describe the expected development timeline is a great idea that again lends itself to adding another illustrative figure, as you suggested.
>
> Finally, we fixed all minor problems; thanks for your careful review.

---

### Official Review · Reviewer_nPP4 · 2025-03-10

**Significance:** 2
**Argument Clarity:** 2
**Rating:** 2
**Confidence:** 4

**Questions:**

1.  What is the specific position this paper is arguing for? Could the authors clearly articulate a thesis statement that distinguishes their view from a general literature review?
2.  Similarly, what unique argument, critique, or new perspective does it introduce that justifies it as a position paper?
3. What major misconceptions, overlooked challenges, or false assumptions in the field does this paper seek to address? Identifying and challenging these would make the paper a stronger position piece.
4.  Could the authors refine their definition of computation in neural cultures to clarify what qualifies as a computational system and what does not?

**Discussion Potential:**

2

**Paper Summary:**

The paper discusses the potential for in vitro neural cultures to serve as computational systems, referencing prior work on neural computation and organoid-based learning. It highlights key experimental challenges, potential methodologies, and the broader implications of in vitro neural computation for AI, neuroscience, and ethics.

**Position:**

Yes

**Position In Title:**

Yes

**Related Work:**

3

**Strengths And Weaknesses:**

Strengths:
- Provides a clear and well-organized overview of in vitro neural computation.
- Summarizes key challenges and opportunities in the field.
- Engages with multiple perspectives from neuroscience, AI, and philosophy.
- Raises important ethical and practical considerations.

Weaknesses
-  The paper nominally presents a position advocating for the importance of testing neural computation in vitro. However, rather than strongly advocating a novel or controversial stance, it largely reviews existing ideas and literature, making it read more like a survey or review paper than a well-defined position paper.
- While the title implies an argument for in vitro neural computation testing, the paper does not present a particularly strong or novel argument, instead summarizing past and ongoing work in this space. Thus, rather than presenting a specific and well-argued position, the paper primarily reviews the state of the field, cataloging various existing perspectives without committing to a strong, well-reasoned stance of its own.
- Overall, the paper does not take a clear, debatable position—it functions more as a literature review rather than as a position paper that advocates for a specific viewpoint. Thus, it does not present original arguments or a strong call to action but primarily compiles and summarizes existing discussions without introducing significant new insights or critiques.  The lack of clear claims, counterarguments, and a structured defense of a single perspective makes the paper less compelling as a position paper.
- Several key concepts, such as "computation" in biological systems, are discussed without a precise operational definition, leaving ambiguity in the argument.

**Support:**

3

---

> ### Author Rebuttal · Authors · 2025-04-01
>
> Thank you for your thoughtful review and feedback! We are pleased that you found our overview clear and well-organized, engaging multiple perspectives and raising important considerations. We address your comments and provide further clarification below.
>
> **Question 1: What is the specific position this paper is arguing for? Could the authors clearly articulate a thesis statement that distinguishes their view from a general literature review?**
>
> We argue that learning to *engineer* useful living neural systems is a practical test to demonstrate and advance our understanding of neural systems (including the artificial neural networks that would be employed in such an effort; see L105). For this to be true, there needs to be a technological capability to engineer in vitro neural systems in the first place. In this context, our more general literature review should be seen as an argument to establish the key premise that technology has indeed matured to the point where the engineering of living neural networks is conceivable.
>
> **Question 2: Similarly, what unique argument, critique, or new perspective does it introduce that justifies it as a position paper?**
>
> The above position may seem unremarkable given that the common machine learning research paradigm has mainly been doing exactly this: engineering artificial neural networks to do practically useful things, with the theoretical or conceptual understanding of resulting systems mostly playing catch-up. But it is important to recognize how the neuroscientific quest is typically approached differently: searching to make sense of given, incredibly complex neural systems. What this position paper seeks to offer as a new perspective is that due to the bioengineering advancements, it may now be possible to extend ML's *engineering-driven* paradigm into the domain of neuroscience. Notably, it may be possible to make practical engineering progress as measured by system usefulness, with theoretical or conceptual understanding following behind. Until very recently, this was arguably technologically infeasible and may still be today, as we discuss in Alternative Views 4.2.
>
> **Question 3: What major misconceptions, overlooked challenges, or false assumptions in the field does this paper seek to address? Identifying and challenging these would make the paper a stronger position piece.**
>
> Thank you for this valuable suggestion. It is possible to frame our argument more in terms of overlooked challenges. In particular, we discuss an often overlooked problem with ML for neuroscience, namely that effective ML-based modeling of observed neural data does not guarantee meaningful insights as long as alternative models and explanations remain equally plausible (see L097). Our proposed engineering-driven approach may offer a way to sidestep this problem as the model can be judged by the task it enables to solve. This challenges a common notion that data collection and algorithmic analysis of in vitro neural activity in itself may be of limited value as long as a formal framework for its interpretation is lacking (Lazebnik, 2004) (see L088). We will update the manuscript to identify and challenge these assumptions more clearly, as suggested.
>
> **Question 4: Could the authors refine their definition of computation in neural cultures to clarify what qualifies as a computational system and what does not?**
>
> Thanks for raising this; we recognize the need for refinement to clarify our definitions.
>
> We adopt a two-level complementary view of computation in neural cultures. At the mechanistic level, computation refers to the systematic transformation of information encoded in neural activity patterns, encompassing the encoding, transformation, and decoding processes (as shown in Figure 1). At the functional level, computation is characterized by the specific information processing task performed by the in vitro system (e.g., MNIST classification).
>
> This allows for distinguishing computational from noncomputational neural activity since performing a task implies the ability for consistent input-output mappings, integration of multiple information streams into coherent outputs, and context-dependent adaptation of processing mechanisms in response to environmental or temporal cues. Random or unstructured neural activity, while potentially involving lower-level processes (e.g., dendritic integration), would not meet these criteria. This is informed by current technological capabilities and may evolve to incorporate new insights into the mechanisms and functions of neural activity. For example, spontaneous "noisy" activity with emergent patterns may qualify as computation if advancing experimental techniques reveal their function. We will clarify these nuances in our revised manuscript to present examples of neural activities that fall within or outside the scope of our definition.
>
> **Weaknesses**
>
> Thanks for this constructive feedback, it greatly helps to revise our clarity.

---

### Official Review · Reviewer_nUv5 · 2025-03-14

**Significance:** 3
**Argument Clarity:** 1
**Rating:** 4
**Confidence:** 4

**Questions:**

What audience is this aimed at? Is this to prove to neuroscientists or to machine learning researchers? If both, what message is for each?

At times the authors make reference to in vitro being a somewhat new technology, yet neuroscientists have used it for 100 years (https://link.springer.com/chapter/10.1007/978-3-030-11135-9_1 . Are they thinking specifically of stem-cell derived cultures?

What is the paragraph about mortality trying to say? I understood very little if it.

**Discussion Potential:**

3

**Paper Summary:**

The paper argues that more focus should be put on the use of in vitro neural cultures. The authors discuss the methodology behind stem-cell based in vitro methods, along with means of recording from and manipulating neurons. The goal is to create hybrid artificial/real neural systems, optimized to perform some task or produce desired outcomes. Such systems have recently been built and the argument here is that they have good properties for studying neural systems, and that this could also help build better AI.

**Position:**

Yes

**Position In Title:**

Yes

**Related Work:**

3

**Strengths And Weaknesses:**

strengths:

-introduces a novel topic, especially for an ML crowd

-offers good counter-arguments

weakness:

-the specific position being advocated for is muddled. Many different connections are made, making it hard to isolate what the core idea is here.

-Writing is vague at times.

-The authors claim that finding ways to control neural networks in vitro will inevitable help us understand brains, but I don't see any evidence of that, or even much space in the article laying out the stops that would lead to that.

**Support:**

3

---

> ### Author Rebuttal · Authors · 2025-04-01
>
> Thank you for your valuable review and feedback! We appreciate your recognition that the paper introduces a novel topic to an ML-focused audience, including good counterarguments.
>
> In the following, we address your comments and provide further clarification.
>
> **Weakness: The specific position is muddled and the writing vague at times**
>
> Please refer to our answer to questions (1-2) of Reviewer nPP4 below.
>
> **Weakness: Claim that finding ways to control neural networks in vitro will inevitable help us understand brains, but I don't see any evidence of that, or even much space in the article laying out the stops that would lead to that**
>
> Thank you for pointing this out. We agree and do not intend to make a claim that the in vitro system will inevitablely help in understanding the brain. Here, we aim to focus on lab-grown neural cultures and argue for the much narrower goal of practical understanding of in vitro systems in the sense of being able to control their neural dynamics. You are correct that this could be achieved without a comprehensive or deeper theoretical understanding of much more complex neural systems such as the brain. We will clarify the wording in this regard.
>
> **Question 1: What audience is this aimed at?**
>
> Our main goal is to reach machine learning researchers, particularly those who may already have expertise or interest in biological or neuro-inspired AI. Our message is to consider extending beyond the common purely simulation-based research (e.g., SNNs/neuromorphic computing). As we argue, this is now becoming possible thanks to stem-cell-derived in vitro neural networks in cloud computing environments, a technological possibility that may not be widely known outside of neuroscientific circles.
>
> **Question 2: Novelty of in vitro technology**
>
> Thanks for this observation! Indeed, when we discuss new in vitro technology, we specifically mean the recent development of stem cell-derived cultures that can now be accessed in cloud computing environments. We will clarify the manuscript in this regard, as you rightly point out that in vitro experiments in general are an older, established capability.
>
> **Question 3: What is the paragraph about mortality trying to say?**
>
> Thanks for this feedback. Our description of Hinton's "mortal computing" terminology may be lacking clarity, and we will seek to improve this in the manuscript's revision. The important idea is that conventional computers have a hardware-software separation such that one can copy software or weights to a new hardware substrate, making the software "immortal". However, achieving this portability requires, by definition, a layer of abstraction, which costs energy. Hinton suggests that if we give up on software portability/immortality (like in the case of mortal in vitro neurons), we may be able to realize more energy-efficient, abstraction-less systems.

---

> > ### Comment · Reviewer_nUv5 · 2025-04-03
> >
> > I thank the authors for their response. It helps me to understand the motivation for this work better. I am willing to increase my score assuming the authors commit to a significant re-framing of the article to make it clearer that this is targeted toward people who are already using spiking neural networks and working in neuromorphic computing. Conveying that this is more about pursuing a new line of engineering, rather than attempting to understand the brain, is helpful (and I believe a more accurate reflection of the author's goals).

---

> > > ### Author Response · Authors · 2025-04-03
> > >
> > > Thank you for your feedback! We greatly appreciate it, as it helps improve the clarity of our paper.
> > >
> > > We are committed to reframing as suggested to better reflect the article's goals. Specifically, we will make significant revisions to more clearly establish that:
> > >
> > > - it is targeted toward machine learning researchers and practitioners, particularly those already working with spiking neural networks and in the neuromorphic computing domain. We note that this was touched upon in Section L133, and we agree that this requires greater prominence and clearer articulation throughout the paper.
> > >
> > > - the primary goal is advancing a new line of engineering within NeuroAI rather than making claims about neuroscientific understanding of the brain. We will explicitly state these boundaries in our introduction and throughout the paper to prevent misinterpretation of our objectives.
> > >
> > > We appreciate your willingness to reconsider your evaluation based on this reframing, as it indeed more accurately reflects our intentions. We believe that these changes will strengthen the paper by providing a clearer context for the readers and positioning our work appropriately within the field.

---

### Decision · Program_Chairs · 2025-04-30

**Decision:**

Accept (poster)

**Comment:**

The reviewers found the position highly relevant and timely. The initial submission came across as a bit "muddled", i.e., it read a bit too much like an overview paper rather than defining and defending a clear position. This was partially caused by an unclear target audience and some unclear terminology and concepts/technology.
The to-the-point author replies nicely managed to address the most important concerns of the reviewers, i.e., re-reading the paper with the additional provided information in mind it becomes a very solid position paper. The authors seem to have nicely grasped what the reviewers were stumbling over, so I am confident they'll manage to make the required changes.